# Environmental diversity of *Candidatus* Babelota and their relationships with protists

Louis Weisse,[1] Lucile Martin,[1] Bouziane Moumen,[1] Yann Héchard,[1] Vincent Delafont[1]

**ABSTRACT** *Ca*. Babelota is a phylum of strictly intracellular bacteria whose representatives are commonly detected in various environments through metagenomics, though their presence, ecology, and biology have never been addressed so far. As a group of strict intracellular, we hypothesize that their presence, occurrence, and abundance heavily depend on their hosts, which are known as heterotrophic protists, based on few described isolates. Here, we conducted a sampling campaign allowing to characterize protists and associated bacterial communities, using high-throughput sequencing. In parallel, a systematic enrichment of protists from samples was performed to attempt characterization and isolation of new *Ca*. Babelota within native hosts. We found that *Ca*. Babelota are among the most widespread phylum among the rare ones. Protist enrichments are allowed in certain cases to enrich as well for *Ca*. Babelota, which could be visualized *in vivo* infecting protist cells. Though cosmopolitan, *Ca*. Babelota diversity was highly site-specific. Cooccurrence analyses allowed to retrieve well-known as well as new putative associations involving numerous protists of various trophic regimes. The combination of approaches developed in this study enhances our understanding of *Ca*. Babelota ecology and biology, while paving the way for future isolation of new members of this elusive phylum, which could have huge impact on protists—and ecosystems—functioning.

**IMPORTANCE** Our understanding of microbial diversity surrounding us and colonizing the environment has been dramatically impacted by the advent of DNA-based analyses. Such progress helped shine a new light on numerous lineages of yet-to-be-characterized microbes, whose ecology and biology are basically unknown. Among those uncharacterized clades is the *Candidatus* Babelota, a bacterial phylum for which parasitism seems to be an ancestral trait. All known *Ca*. Babelota thrive by infecting phagotrophic protist hosts, thereby impacting this basal link of the trophic chain. The *Ca*. Babelota constitutes a model that stands out, as phylum-wide conserved parasitism has only been described in one previous occurrence for Bacteria, with the Chlamydiota. Thus, exploring the intricate interplay between *Ca*. Babelota and their protist hosts will advance our knowledge of bacterial diversity, their ecology, and global impact on ecosystem functioning.

**KEYWORDS** symbiosis, host-microbe interactions, protists, intracellular bacteria

Bacteria are richly present in every environment, in interactions with other (micro)organisms, such as protists, viruses, or other bacteria (1). Despite their overabundance in the environment, a multitude of bacteria is yet uncultivable, a phenomenon first witnessed and formalized with the concept known as the great plate count anomaly (2). Our current view on microbial diversity clearly underscores that the vast majority of the bacterial domain remains elusive and resist cultivation attempts (3, 4). Collectively, those uncultivated bacteria can be referred to as the microbial dark

**Peer Reviewer** Qinglin Chen, University of Melbourne, Melbourne, Australia

Address correspondence to Vincent Delafont, vincent.delafont@univ-poitiers.fr.

The authors declare no conflict of interest.

See the funding table on p. 15.

matter (MDM) (5). Most notably, the recent description of the Candidate phyla radiation (CPR), a group representing 15% of the known bacterial diversity, emphasized how much there is left to discover and understand of the diversity and functioning of MDM (6). Subsequent analyses highlighted that CPR members almost systematically present small sizes as well as genomes, lacking numerous biosynthetic pathways and thus testifying of reliance toward other (micro)organisms for its development (7). Such associations, which could be considered symbioses *sensu* de Barry, were repeatedly described with several MDM clades such as *Candidatus* Parcubacteria, which can be found in endosymbiosis with *Paramecium bursaria*, or *Candidatus* Saccharibacteria that lives in an epibiotic parasitic symbiosis attached to *Actinobacterium*, within the oral microbiota (8–11).

*Candidatus* Babelota is a phylum related to the CPR, first described in 1996 as TM6 clade (12). Later, the in-depth analysis of the first *Ca*. Babelota genome suggested this phylum exclusively comprises bacteria adopting an intracellular lifestyle (13). All *Ca*. Babelota genomes show marked size reduction along with AT bias, further substantiating that the intracellular lifestyle is a defining trait of this phylum (14). For this reason, the phylum was then provisionally renamed *Ca*. Dependentiae (15). Recently, with the aim to harmonize bacterial taxa nomenclature, this phylum was renamed as *Ca*. Babelota (16, 17). The presence of *Ca*. Babelota was highlighted in multiple studies, always by indirect means of detection, such as high-throughput sequencing (14). Their omnipresence was demonstrated in multiple environments, but always in low relative abundance and almost never being discussed. A feature standing out is the fact that all *Ca*. Babelota isolates currently maintained in laboratory settings are infecting protists, which are thus the sole type of host cells characterized for this phylum (18–21).

So far, members of the Amoebozoa (*Acanthamoeba* and *Vermamoeba*) and Stramenopiles (*Spumella*) are known to host *Ca*. Babelota. Recently, a single-cell metagenomics approach further added other amoebae (the genus of testate amoeba *Hyalosphenia*) and the alveolate ciliate *Loxodes* sp. as host for *Ca*. Babelota (22). Additionally, it appears that habitats harboring the highest occurrence, abundance, and diversity for *Ca*. Babelota (based on *in silico* predictions) match those of numerous phagotrophic protists, e.g., freshwater sediments, soil, among others (14, 23). Protists are constituting the vast bulk eukaryotes diversity, present in virtually all environments and playing important parts in ecosystems functioning (24). They harbor different trophic regimes, ranging from phototrophy to heterotrophy, therefore, representing an important part of the food chain (25). Indeed, they are also involved in a multitude of other interactions, either in symbiosis with other organisms such as plants or animals or by hosting prokaryotes (26, 27). However, although symbioses between protists and prokaryotes are thought to involve virtually all major protist lineages, the current knowledge on symbioses diversity may be strongly biased toward easily cultivable protists, such as amoebozoans and ciliates, for example; a bias possibly witnessed with *Ca*. Babelota (14, 27). Based on this current knowledge, we, thus, hypothesize that *Ca*. Babelota constitutes a phylum of strict intracellular that could stably infect a vast array of protists, not only restricted to previously described associations such as with the Amoebozoa and Stramenopiles.

To advance our understanding of *Ca*. Babelota ecology, and their interactions with various protist hosts, we designed a soil, water, and sediment sampling scheme to investigate environmental niches for both *Ca*. Babelota and protists. For each environmental sample, we conducted empirical protist enrichments alongside systematic high-throughput sequencing of both prokaryotic and microeukaryotic communities. The combination of benchwork and molecular biology allowed us to investigate the intricate relationships between the *Ca*. Babelota and their putative protist hosts, through direct observations from cultures and cooccurrences analyses.

## MATERIAL AND METHODS

### Sampling

Samples were collected from April 2022 to February 2023 in Nouvelle-Aquitaine region, France. In total, 14 sites were sampled four times per year (Fig. 1; Table S1). Water samples (2L) were collected using a deep sampler (Grosseron, ref. P415114), targeting the bottom 30 cm of the water column. Sediment samples were collected from the upper layer (top 10 cm) using 50 mL sterile tubes. Soil samples (top 10 cm) were collected using a sterile spatula. All samples were stored at 4°C after collection until processing, up to 72 h post sampling. Temperature, salinity/conductivity dissolved oxygen, and RedOx potential were measured onsite using a multiparameter water probe (Hanna Instrument, ref. HI98494). For soil and sediment samples, humidity percentage was determined by weighing the sample before and after 3 days drying at room temperature. pH was measured by mixing 5 g of dried soil with 25 mL of distilled water.

### Phagotrophic protist enrichment

Water samples were filtered on 3 µm mixed cellulose esters filters (Millipore, ref. SSWP04700) then scraped in 5 mL of Page amoeba saline solution (PAS; 4 mM $MgSO_4$, 0.4 M $CaCl_2$, 0.1% sodium citrate dehydrate, 2.5 mM $NaH_2PO_3$, 2.5 mM $K_2HPO_3$, pH 6.5),

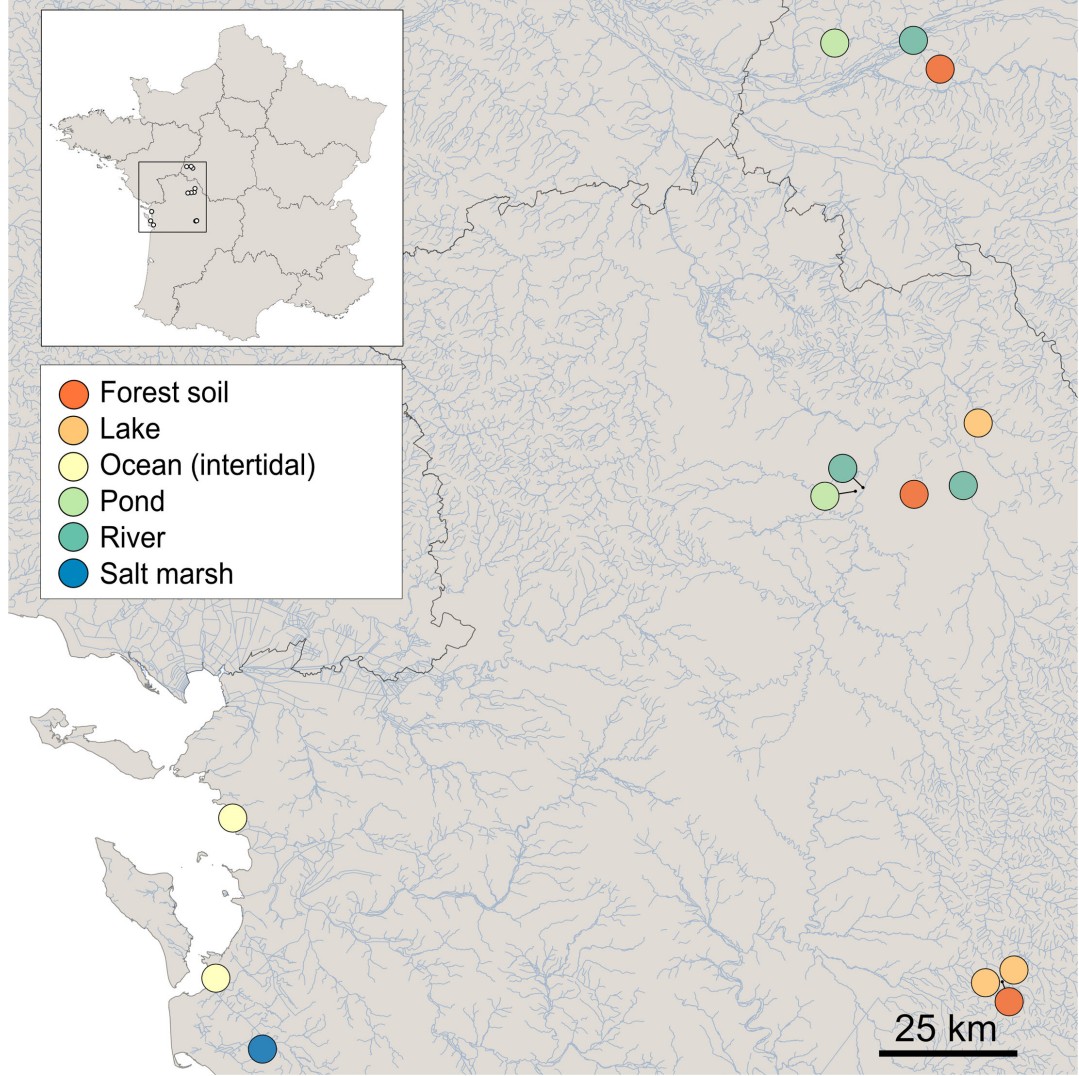

**FIG 1** Geographic locations of selected environments for sampling, loated in Nouvelle Aquitaine region, France. Each environmental category is color-coded.

and before being transferred into flasks. Soil and sediment samples were enriched by inoculating 1 g of particles sieved with a 500 µm cutoff, in 7 mL of PAS. The suspensions were supplemented with one of two food sources: (i) either addition of *Escherichia coli* MG1655::ΔtolC (final concentration 5·10$^8$ bacteria/mL (28) or (ii) with a 1 cm$^2$ piece of yeast malt agar (55 mM dextrose, 0.5% peptone, 0.3% malt extract, 0.3% yeast extract, 2% agar, pH 6.2), as an optimized protocol from previously described media for protist culture (29).

Flasks were incubated at 20°C in the dark and subcultured weekly. For each subculture, flasks were vigorously tapped to ensure detachment of adherent cells, and 500 µL from the grown flask was transferred to a new flask. The volume was then adjusted to 7 mL with PAS and supplemented with the corresponding food source.

## Fluorescence *in situ* hybridization

After resuspending the cells by tapping the flasks, 2 mL of each sample was transferred into a sterile mL tube and centrifuged at 5,000 × *g*, for 5 min to pellet the cells. The supernatant was discarded, and pellets were resuspended in 400 µL of 4% paraformaldehyde followed by a fixation for 30 min at room temperature in the dark. The fixed samples were then centrifuged at 1,500 × *g* for 5 min. Supernatants were discarded, and pellets washed twice with PAS buffer and finally resuspended in 200 µL of 50% ethanol.

Fluorescence *in situ* hybridization (FISH) was performed according to the standard FISH protocol v2.2 available in the SILVA database website (https://www.arb-silva.de/fish-probes/fish-protocols/)(30). Hybridization was carried out using a combination of probes EUB338 I-III (5′-GCTGCCTCCCGTAGGAGT-3′, 5′-GCAGCCACCCGTAGGTGT-3′, 5′-GCTGCCACCCGTAGGTGT-3′) and EUK516 (5′-ACCAGACTTGCCCTCC-3′) (31, 32). Additionally, a probe specifically targeting *Ca*. Babelota was designed and used in this study (TM6_681: 5′-GCATTTTACCGCTACTCC-3′) (File S1). Probes targeting bacteria were doubly labeled with fluorochromes, to enhance fluorescence intensity and rRNA accessibility (33). Hybridization was conducted with 15% formamide, for 3 h at 46°C. Rinsing was done using washing buffer with adjusted salt concentrations, for 25 min at 48°C. The slides were then rinsed with cold distilled water, air dried, and mounted using Fluoroshield with DAPI before observations. Slides were observed using Olympus IX73 epifluorescence microscope and Olympus Fluoview FV3000 confocal laser scanning microscope. Merging of fluorescence channels was done in ImageJ (34).

## DNA extraction

DNA extractions were done using the QIAcube Connect (QIAGEN). Nucleic acids from environmental samples were extracted using the DNeasy PowerWater Kit (14900-100-NF) for water samples after filtration on 3 µm filters (Millipore, ref. SSWP04700) and with the DNeasy PowerSoil Pro Kit (47014) for the soil samples according to the manufacturer's instructions. For protist enrichment cultures, nucleic acids were extracted using DNeasy Blood & Tissue Kit (69504) as per the manufacturer's protocol. This involved scraping all organisms from the flask, centrifugation at 5,000 × *g* for 5 min, and removal of the supernatant.

## Quantitative PCR

To assess the presence of *Ca*. Babelota in both environmental samples and protist enrichments, quantitative PCR was conducted. A new pair of primer was designed, based on an alignment of full-length (>1,400 nt) 16S rRNA sequences affiliated to the *Ca*. Babelota phylum in Silva database (accessed on October 2017). This alignment served for primer design using PrimerDesignM software (35). A primer set flanking the V5 variable region (TM6-V5-F2, 5′-GGAGTAGCGGTAAAATGC-3′; TM6-V5-R4, 5′-CTACCAGGGTATCTAATCC-3′) was selected, tested, and validated both *in silico* and *in vitro* (File S1).

qPCR mixtures were prepared in duplicate with 250 nM of forward and reverse primers, mixed with 5 µL of Takyon No ROX SYBR 2X MasterMix blue dTTP (Eurogentec

Ref. UF-NSMT-B0701), 1 µL of DNA template and 3.5 µL of nuclease-free distilled water. Amplification was performed with a LightCycler 480 System (Roche Diagnostics Ref. 05015278001), applying an initial denaturation for 5 min at 95°C followed by 45 cycles of denaturation for 10 s at 95°C, annealing for 10 s at 60°C, and elongation for 10 s at 72°C. A fusion curve was produced by denaturing samples for 5 s at 95°C, and then a gradual temperature rises from 65°C to 95°C at a rate of 0.11°C/s including five fluorescence acquisitions per °C.

## DNA sequencing

For bacterial communities, amplicons were prepared using the Quick-16S Plus NGS Library Prep Kit (V3-V4) (Zymo Research, D6420). This kit is based on the use of primers 341F (5′-CCTACGGGDGGCWGCAG-3′ and 5′-CCTAYGGGGYGCWGCAG-3′) and 806R (5′-GACTACNVGGGTMTCTAATCC-3′) targeting the 16S rRNA V3-V4 region and combining the barcoding step with a qPCR to quantify gene copy numbers. All steps were performed according to the manufacturer's instructions. DNA samples were then processed for sequencing using Miseq System (Illumina) for $2 \times 300$ cycles. This work benefited from equipment and services from the iGenSeq core facility, at ICM, France.

For protist communities, amplicons were generated using primers 18S-EUK581-F (5′-GTGCCAGCAGCCGCG-3′) and 18S-EUK1134-R (5′-TTTAAGTTTCAGCCTTGCG-3′), designed to target protists while excluding metazoans (36). The Q5 high fidelity polymerase (New England Biolabs) was used for all PCR assays according to previously established protocols and guidelines (37, 38). PCR mixtures were prepared in duplicate according to the following: 10 µL of 5× buffer, 1 µL of dNTP (200 nM each), 250 nM of both forward and reverse primers, 0.5 µL of Q5 polymerase (1 U), 28.5 µL of nuclease-free water, and 5 µL of DNA template. Amplification was performed by applying an initial denaturation step for 30 s at 98°C, followed by 30 cycles with denaturation at 98°C for 10 s, annealing at 51.1°C for 20 s, and elongation at 72°C for 45 s with a final elongation also at 72°C for 2 min. Amplicons were checked on 1% agarose gel and purified using the QIAquick PCR Purification Kit (QIAGEN, Ref. 28106). A pool of amplicons (200 fmol each) was used for library preparation using the Native Barcoding Kit V14 (Oxford Nanopore, Ref. SQK-NBD114.96) and following the manufacturer's protocol. Libraries were loaded on R.10.4.1 flowcells and sequenced for 24 h on high accuracy mode using a MinION Mk1C sequencing device (Oxford Nanopore, Ref. MIN-101C).

## Bioinformatic analyses

For Illumina sequence data, demultiplexing was performed at the IGenSeq core facility, and the resulting fastq files were analysed using the QIIME2 software package (39). Quality control included trimming and truncating primer sequences, removing low-quality ends ($Q < 30$), denoising, merging reads, and checking for chimeras using DADA2 within QIIME2 (40). Amplicon sequence variants (ASV) obtained from DADA2 were further curated by filtering out singletons. The remaining ASVs were used to infer taxonomy, using a naïve Bayes classifier, trained on V3-V4 trimmed 16S sequences, along with the SILVA v138 database. An additional filtration step was applied to remove chloroplastic and mitochondrial sequences. Alpha and beta diversity measures were done in QIIME2 and using the vegan package in R environment (41).

For Oxford Nanopore sequence, the raw fast5 files were basecalled, demultiplexed, and quality controlled (based on Q9 cutoff) with Guppy v 6.4.6. The resulting FASTQ files were then used for inferring taxonomy, using emu v3.4.5 along with the protist ribosomal reference database v5.0.1 (42, 43). Relative abundance data files were further explored in R environment using ggplot2 and vegan packages (41, 44).

Interkingdom cooccurrence analyses were performed using the cooccur package v 1.3c (45). Only cooccurrences showing p_gt < 0.05 were considered for further analyses. Briefly, the p_gt value corresponds to the probability of co-occurrence between two species, at a frequency greater than the expected one. Values < 0.05 for p_gt, thus, reflect

statistically significant positive cooccurrences. All generated networks were visualized in Cytoscape v 3.10.1 (46).

## Phylogenetic analyses

Full length 16S rRNA *Ca*. Babelota sequences were retrieved from the SILVA database (accessed in April 2024), aligned using MUSCLE v3.8.31 (47). The resulting alignment was manually curated and cleaned using BMGE v1.12 (48). Phylogeny was inferred using IQ-tree v2 (49). ModelFinder, implemented in IQ-tree, was used to find the best fit substitution model (50). Maximum likelihood phylogeny was reconstructed, and node robustness was assessed using 1000 iterations of conventional bootstraps and Shimo-daira–Hasegawa approximate likelihood ratio tests.

## RESULTS

### *Ca*. Babelota are commonly found in diverse environments, within protists

A previous meta-analysis of amplicon sequence (meta)data, identified putative hotspots for *Ca*. Babelota presence (14). Leveraging this, we collected samples from the various identified environmental hotspots throughout one year (Fig. 1). Overall, 14 sites were sampled once during each meteorological season, yielding a total of 97 samples.

A specific qPCR approach identified the presence of *Ca*. Babelota in 55 out of 97 (56.7%) samples (Fig. S1). This sampling scheme allowed us to examine the impact of various factors on the presence and abundance of *Ca. Babelota* in the environment (Fig. 2A). Seasonality emerged as the most statistically significant factor affecting *Ca*. Babelota, with spring seemingly being the least favorable season for these bacteria.

Given our hypothesis that *Ca*. Babelota depend on phagotrophic protist hosts for their lifecycle, we implemented a systematic enrichment of protists from samples. Out of the 97 screened samples, we observed an increase in bacterial concentration (i.e., enrichment in at least 2 consecutive subcultures) in 14 environmental samples (Fig. 2B). In two enrichment occurrences, *Ca*. Babelota were observed using fluorescent *in situ* hybridization (FISH). In those cases, *Ca*. Babelota were systematically located within the cytoplasm of microeukaryotes (Fig. 2C; Fig. S2). Among all protist enrichments, sequences analyses indicated that all positive samples fell within 4 *Ca*. Babelota ASV, affiliated either to the Chromulinavoraceae (2 ASV) or Vermiphilaceae (2 ASV) families. Among those, 1 was successfully maintained in culture for 11 weekly subcultures. Sequence analyses revealed an enrichment of a single *Ca*. Babelota 16S rRNA sequence, which was similar to *C. destructans* (424 nucleotides, 100% identity). The enriched protist communities in those samples varied according to the samples though it showed a systematic increase in various phagotrophs (Table S2).

### Quantitative 16S rRNA sequencing highlights prokaryotic environmental richness driven by rare phyla

High-throughput sequencing (HTS) of 16S rRNA amplicons was also conducted on all 97 samples. From this data set (average of 92,923 ± 30,616 sequences per sample), 11 archaeal and 73 bacteria phyla were identified. Bacterial sequences dominated (96.3%) with high abundances of Pseudomonadota, Cyanobacteriota, Bacteroidota, Actinobacteriota, Acidobacteriota, and Desulfobacteriota. Altogether those 6 phyla represented 82% of all generated sequences (Fig. 3A). Global trends in abundances of certain phyla were observed, such as higher abundances of Actinobacteriota and Acidobacteriota in forest soil samples, compared to all other categories. Similarly, Cyanobacteriota were mostly observed in lake and salt marsh samples. In contrast, 49 out of the 73 identified bacterial phyla each represented less than 0.1% of the total bacterial data set (Fig. 3B). Alpha-diversity measures highlighted a global trend of higher richness in soil and sediment samples, compared to water samples (Fig. 3C). The lowest observed richness was witnessed in salt marsh water, whereas the highest richness (along with evenness) was found in all freshwater sediments (Fig. 3D). On the opposite, marine samples showed

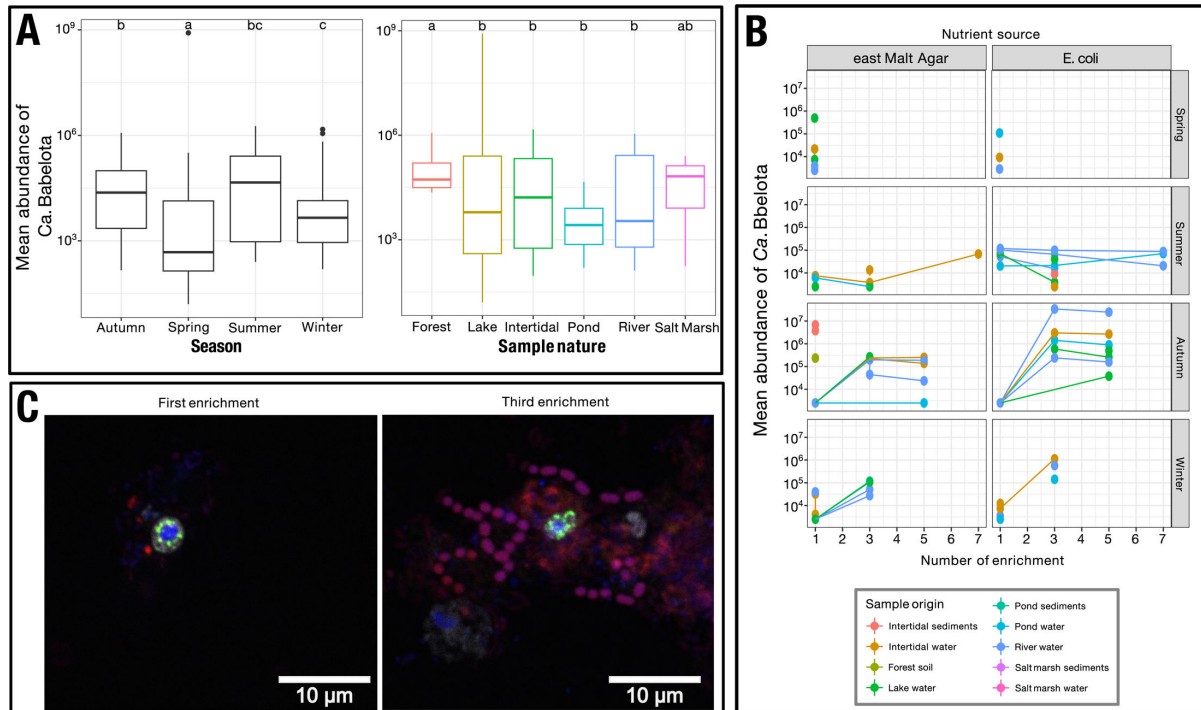

**FIG 2** Candidatus Babelota are detected all year long, in every sampled environment, and their presence can be favored in protist through enrichment. Mean abundance of *Ca*. Babelota detected by quantitative PCR depending on season or sample nature (A). Mean abundance of *Ca*. Babelota detected by quantitative PCR in protist enrichments depending on nutrient food source, season, and subculture advancement (B). Fluorescence *in situ* hybridization imaging selected protist enrichment originally sampled from a pond covered in duckweed (C). Green signal corresponds to probe TM6_681 targeting *Ca*. Babelota. Red signal corresponds to the eubacterial probes EUB338-I-III. White signal corresponds to EUK516 probe targeting eukaryotic cells. DNA was labeled using DAPI, shown in blue.

the lowest evenness. Coupling Illumina HTS library preparation with qPCR allowed estimating prokaryotic concentration in samples, which ranged from $9.75 \times 10^2$ to $1 \times 10^7$ 16S rRNA gene copies per mg of soil or sediment, whereas water samples ranged from $2.53 \times 10^2$ to $6 \times 10^7$ rRNA gene copies per mL (Fig. 3E). Notably, salt marsh water stood out by showing the highest concentration, compared to all other sample categories, though no significant differences were noted (Kruskal-Wallis test, $\chi^2 = 17.34$, df = 10, $P = 0.067$). Exploratory analyses suggested that sample origin and nature, along with pH and salinity/conductivity, are the most statistically significant factors shaping community composition (Fig. 3F). On the opposite, neither temperature, season nor dissolved oxygen showed significant impact.

## *Ca*. Babelota is among the most frequently occurring rare phyla in all environments

Despite the overwhelming predominance of few phyla, most of the observed diversity corresponds to a myriad of lowly abundant phyla (i.e., less than 0.1% of total data sets). Focusing on those, only 13/71 were identified in more than 50% of the samples. Notably, four lowly abundant phyla were occurring in at least 75% of samples that are namely Fibrobacterota, *Ca*. Babelota, Elusimicrobiota, and SAR324 (Fig. 4A). More specifically, *Ca*. Babelota were identified in 85/97 samples (87.6%) and was represented by a total of 3,606 sequences clustered in 354 ASV. *Ca*. Babelota were broadly found across all sample categories, throughout the year (Fig. 4B). Soil and salt marshes stood out as being always positive for *Ca*. Babelota, whereas intertidal water showed the lowest occurrence, i.e., 75% (15/20 samples).

The distribution of *Ca*. Babelota ASV in environmental categories highlighted a site-specific diversity. Out of the 354 *Ca*. Babelota ASV, only 16 were shared among 2 sample

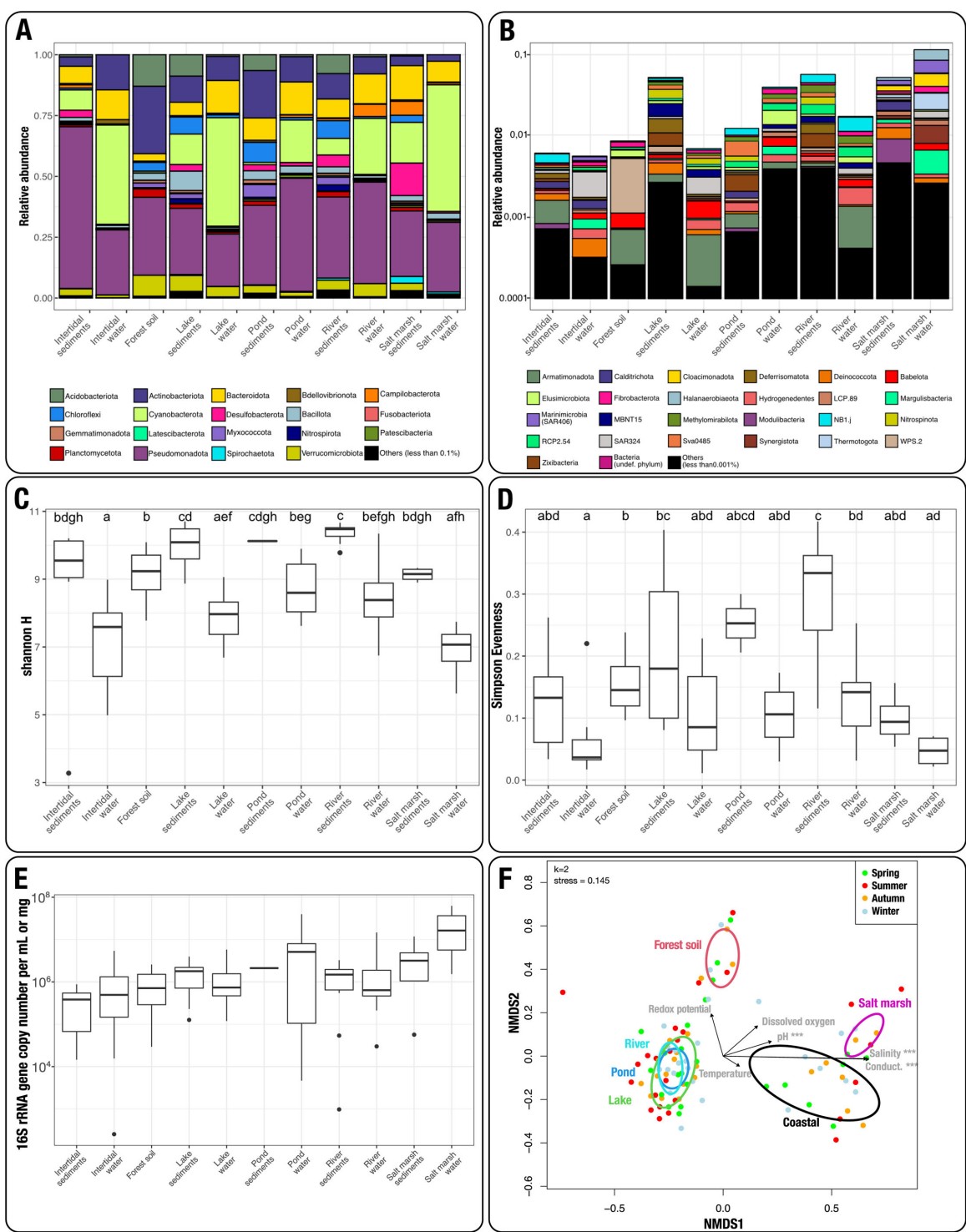

FIG 3 Bacterial communities vary depending on environment and physicochemical parameters and are driven by lowly abundant phyla. Relative abundance of highly represented bacterial phyla (more than 0.1% of each represented data set (A). Relative abundance of lowly represented bacterial phyla (less than a0.1% of each represented data set (B). Shannon's diversity index of bacterial communities by environment (C) and Simpson evenness index of bacterial communities by environment (D). Significant statistical variations in diversity and evenness among groups are represented by letters, as evaluated using Kruskal-Wallis and pairwise comparisons using Wilcoxon rank tests (adjusted with the Benjamini-Hochberg method) tests with $P < 0.05$ threshold. Quantification of 16S rRNA gene copy numbers in all samples according to their environmental category (E). Non-metric multidimensional scaling of bacterial communities displayed using two dimensions (F). Each arrow represents a physicochemical parameter significantly impacting communities (*** indicates $P < 0.001$).

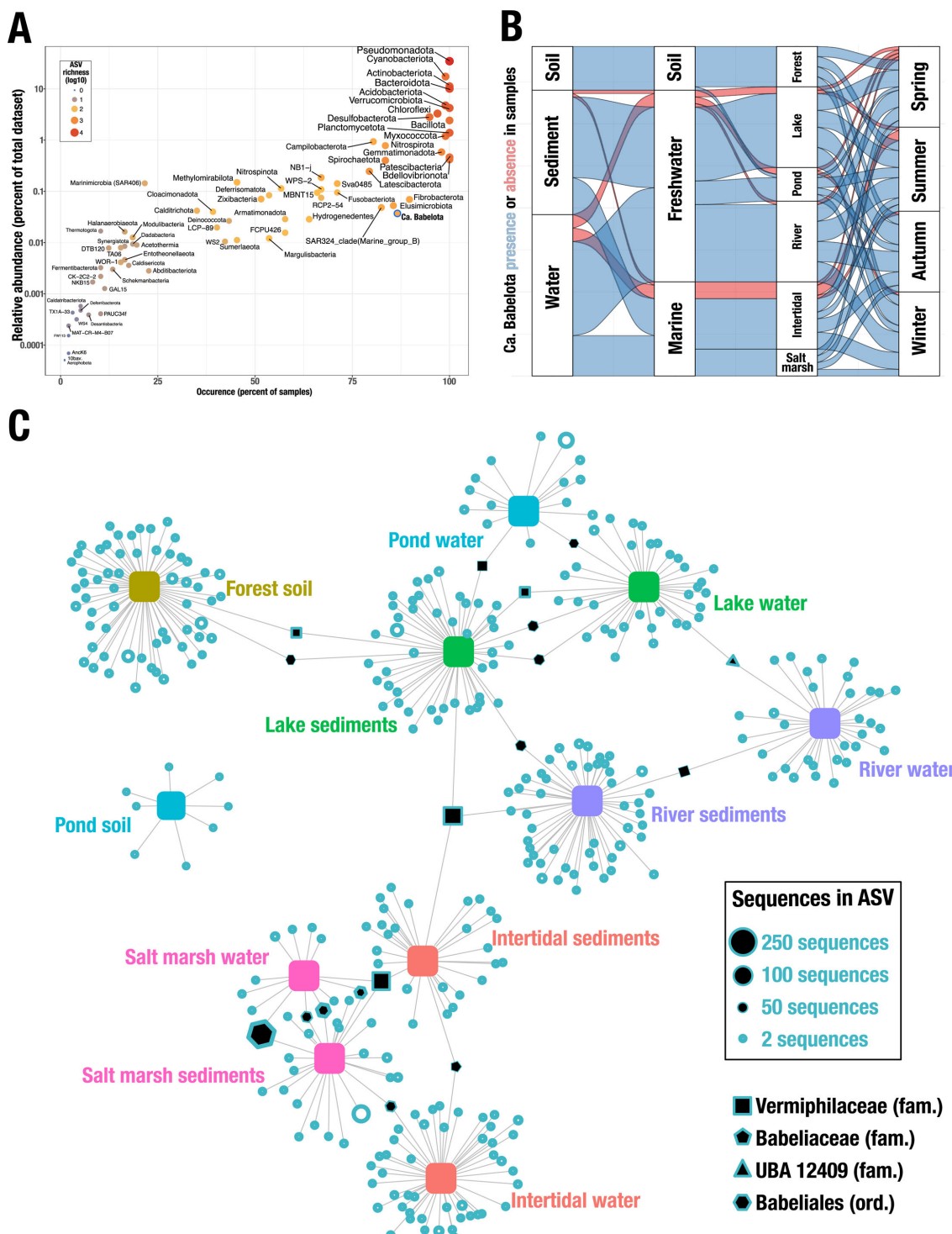

**FIG 4** *Ca.* Babelota is one of the most frequently occurring lowly abundant phyla in all environments, showing a site-specific diversity. Relative abundance as a function of sample occurrence, for all detected bacterial phyla (A). Alluvial graph detailing the occurrence of *Ca.* Babelota in samples according to their various features (B). Blue alluvia reflect *Ca.* Babelota presence in a given category, whereas red alluvia indicate its absence. Network analysis depicting the repartition of all *Ca.* Babelota ASV identified in this study, according to environmental categories (C). Each environmental category is represented by a central, square-shaped node with a defined color. White circles represent a *Ca.* Babelota ASV found in one environment, whereas ASV found in multiple environments are filled black. Size of ASV shapes is proportional to their sequence count, whereas the shapes differentiate the various family-level clades.

categories and 2 shared among 3 sample categories (Fig. 4C). The 2 most cosmopolitan ASV were affiliated with the Vermiphilaceae family; the other 16 ASV fell within either of Vermiphilaceae, Babeliaceae, UBA12409 families, or Babeliales order. Additionally, few *Ca*. Babelota were represented by large sequence counts; 11 were represented by 50–270 sequences, whereas 73 ASV counted only 2 sequences. Indeed, most of the highly represented ASV are those shared in multiple sample categories. However, some lowly represented ASV were shared in two samples categories and vice versa (Fig. 4C; Fig. S3).

The collected *Ca*. Babelota ASV were placed into 16S rRNA gene-based phylogeny, reconstructed using sequences from the SILVA database (2,637 sequences from release 138). The 354 *Ca*. Babelota ASV identified in this study covered all known taxonomic group within the phylum (Fig. 5). Most notably, the Vermiphilaceae family was the taxonomic group encompassing the highest ASV diversity (65 ASV). The Vermiphilaceae, along with the Chromulinavoraceae (10 ASV), were found in all sampled environments. The Babelaceae (3 ASV) were only found in lake, river, and salt marshes. Altogether, most ASV (337/354) were shown to belong to the Babeliae, the sole formally described class, for which genomic data are currently available. Added to that, phylogenetic analyses indicated the presence of another class (named here Babelota class II), represented by 17 ASV identified in the present study.

## Protist diversity analyses help pinpointing putative *Ca*. Babelota hosts

Because we hypothesize that protists are hosts for *Ca*. Babelota, we identified protist communities present in all samples using 18S rRNA amplicons HTS. Among the 97 samples, 2,312,959 eukaryotic sequences were produced, accounting for 22 phylum-level clades after removing mitochondrial, plastidic sequences, as well as sequences originating from pluricellular organisms and bacteria. Cryptophyta were the most represented microorganisms in the data set, displaying the highest relative abundance in 8 out of the 11 environmental categories (Fig. 6A). Along with Cryptophyta, six other protist clades were present in all sample categories, namely, Chlorophyta, Stramenopiles, Rhizaria, Tubulinea, Alveolata, and Discoba (Table S3). Among all our samples, salt marsh waters appeared once again as outsiders, showing the lowest richness and evenness of all environments (Fig. 6B). Despite this observation, no significant differences could be observed among sample categories. To understand dynamics of protist communities, exploratory analyses were conducted to understand the impact of environmental parameters, showing that temperature and pH had strong (significant) effects on protist communities while dissolved oxygen, salinity, and redox potential showed no significant impact (Fig. 6C).

To explore the impact of protists on bacterial communities, relative abundances of protist phyla were used as environmental factors for NMDS analyses based on bacterial populations (Fig. 6D). Among the 22 detected phyla, 10 displayed a significant impact on bacterial communities. Looking further into these 10 phyla, 2 are frequently described as autotrophs (Rhodophyta and Chlorophyta), 5 as heterotrophs (Apusomonoadas, Picozoa, Centroplasthelida, Choanoflagellata, and Perkinsea), and 3 are described as heterotrophic, autotrophic, or *bona fide* mixotrophic regime (Dinoflagellata, Gyrista, and Chrompodellids).

To specifically address putative associations between protists and *Ca*. Babelota, interkingdom cooccurrence analyses were performed, identifying 12 species that cooccurred positively with eukaryotic microorganisms and/or other *Ca*. Babelota (Fig. 6E; Table S4). Rhizaria phylum was involved in the highest number (31%) of observed cooccurrences, followed by Stramenopiles (28%). Only two protist genera stood out as interacting more frequently than others with *Ca*. Babelota species, namely, *Acanthamoeba* (free-living amoeba) and *Prorocentrum* (mixotrophic dinoflagellate). Among all the putative host found, heterotrophy seemed to be the preferential trophic trait, as 64.5% of protists found (20 out of 31) are heterotroph, 25.8% (8 out of 31) are phototroph, and 9.7% (3 out of 31) are mixotroph. Even though the cooccurrence network showed that some *Ca*. Babelota species could interact with multiple other organisms (up

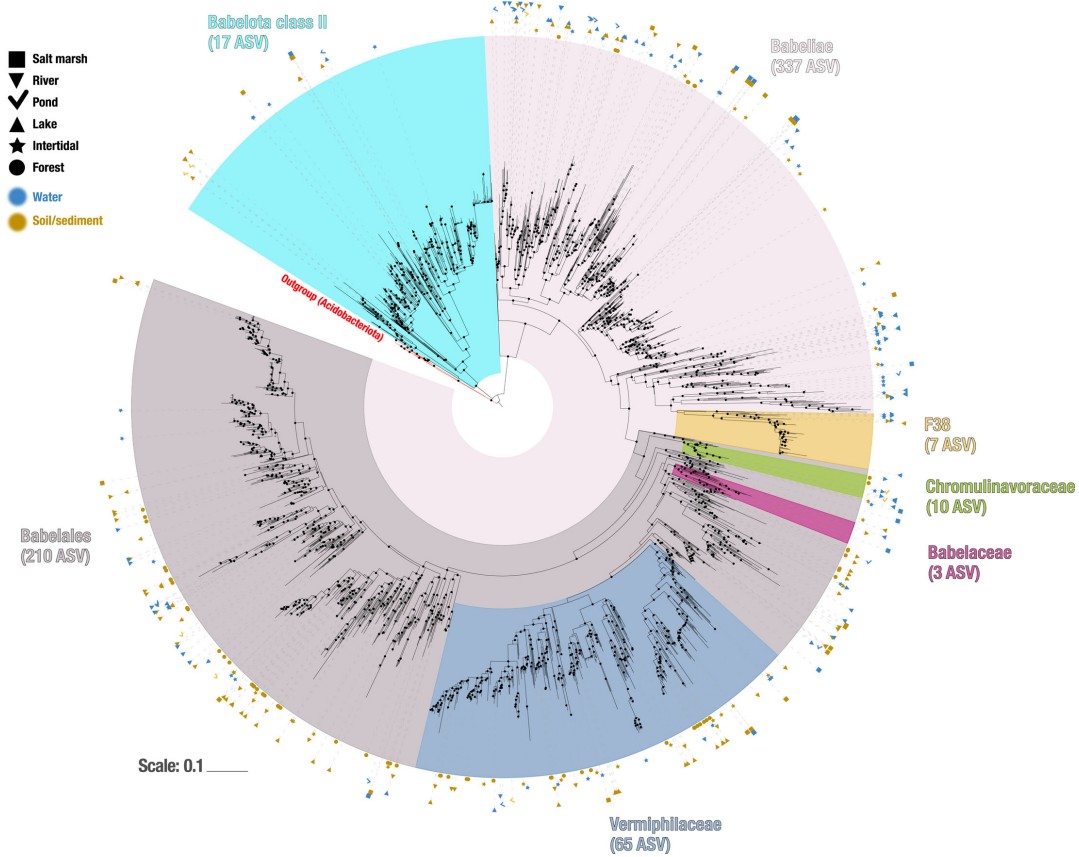

**FIG 5** A 16S rRNA-based phylogeny of *Ca*. Babelota ASV highlights a complete coverage of all known clades within this phylum and highlights the vast diversity and omnipresence of the Vermiphilaceae family. Phylogeny was reconstructed using maximum likelihood, with IQ-tree 2, using the GTR + F + I + G4 model, according to the best Bayesian Information Criterion score obtained in ModelFinder. Sequences from the Acidobacteriota phylum were used as an outgroup for rooting. Node support of at least 75% (for both conventional bootstraps and Shimodaira–Hasegawa approximate likelihood ratio tests) is depicted by black filled circles at nodes. Scale represents 0.1 nucleotidic substitution per site.

to 15, i.e., 12 protists and 3 bacteria), 4 species showed a positive cooccurrence with a single protist taxon, which could provide more precise information for identifying the potential host of these intracellular bacteria. Among the four pairs, the presence of *Vermiphilus pyriformis* was shown in cooccurrence with *Vermamoeba*, a genus of free-living amoeba already known to harbour this bacterium. Indeed, all positive cooccurrences shown in our analysis may reflect yet-unknown putative associations.

## DISCUSSION

*Ca*. Babelota is a bacterial phylum found in a large array of environments worldwide (14). The few *Ca*. Babelota isolates described since 2015 (four in total) have shown they are strictly intracellular, infecting phagotrophic protists. Genomic analyses suggest this intracellular lifestyle is a phylum wide conserved trait (15). These findings prompted us to further investigate *Ca*. Babelota ecology and seek new isolates by examining their putative protist hosts.

To achieve this, a sampling scheme, driven by anterior *in silico* analyses, was implemented to probe *Ca*. Babelota presence across various environments. We focused our sampling effort on particle-associated bacterial diversity, i.e., using a filtration step selecting for microorganisms and particles larger than 3 μm, along with their associated microbial communities. The originality of our approach lies in its effort to biologically explore our hypotheses gathered through sequence-based data, for all samples. To do so, we postulated that empirically enriching for phagotrophic protists could enhance *Ca*.

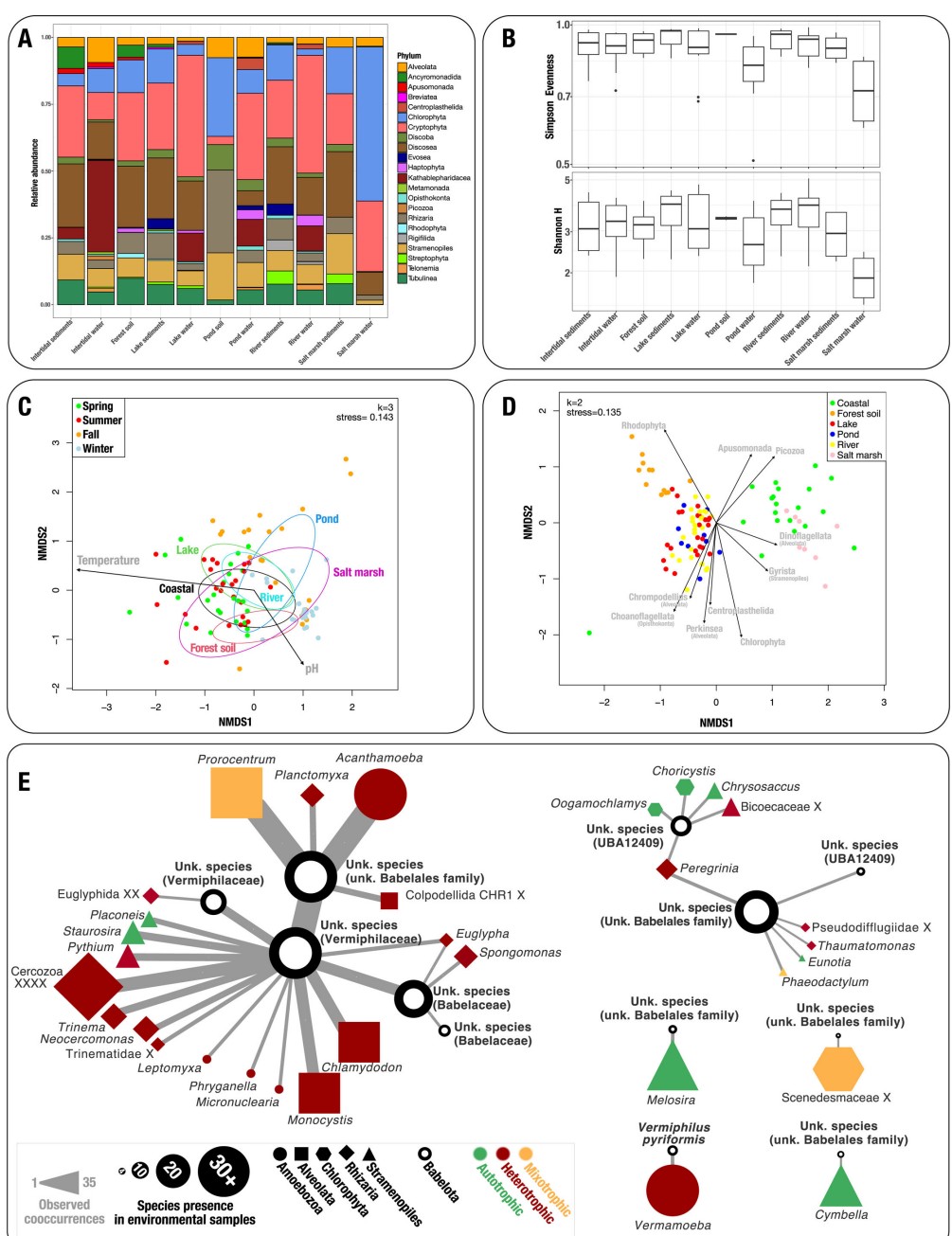

**FIG 6** Protist diversity varies according to their environmental origin and shape bacterial communities, including *Ca*. Babelota. Relative abundance of protist phyla by environmental origin (A). Simpson evenness and Shannon diversity index of protist communities by environment (B). Non-metric multidimensional scaling of protist communities (C). Each arrow represents a physicochemical parameter significantly impacting communities. Non-metric multidimensional scaling (NMDS) of bacterial communities, using protist phyla relative abundance as environmental parameters (D). Each arrow represents a protist clade significantly impacting the bacterial communities. Network of significant (*P* < 0.05) interkingdom positive cooccurrences involving *Ca*. Babelota species (E).

**Babelota recovery.** Previous work has proven that using protist culture can result in the isolation of *Ca*. Babelota (18–20). Our method did not result in the stable maintenance of new *Ca*. Babelota isolates. However, we detected *Ca*. Babelota enrichment in 14 out of 97 protist cultures. Among those positive samples, only ASV from the Vermiphilaceae and Chromulinavoraceae families were enriched. This low rate of success and phylogenetic bias may suggest that some *Ca*. Babelota clades could benefit from an array of host that

spans beyond that of protists, potentially including multicellular organisms, as witnessed for the Chlamydiota, the sole documented phylum of intracellular bacteria (51). Though the success rate appears low, our approach yielded several occurrences of *Ca*. Babelota was maintained in the laboratory and directly observable within eukaryotic cells, thus validating the idea that recovering these bacteria through protist cultivation is feasible. Despite that, numerous improvements could be applied to the method. Most notably, the implementation of cell sorting, coupled with (viable) fluorescent labelling of microorganisms, could dramatically increase success in isolating, identifying, and maintaining protists with their endosymbionts (52–55). The development and validation of a *Ca*. Babelota FISH probe, used in this study, lays the groundwork for employing such methods in future research. A major limitation of our protist enrichment method is its bias towards neutral and positive interactions, potentially overlooking highly parasitic relationships. For example, *Babela massiliensis* and *Chromulinavorax destructans* are both known to induce rapid host lysis. Despite that, the most stable protist enrichment obtained in our study (11 weeks; Fig. 2C) was, in fact, an isolate closely related to *C. destructans*. This observation emphasizes the variability of *Ca*. Babelota interactions phenotypes with their hosts. Overall, pursuing efforts to isolate new *Ca*. Babelota will undoubtedly provide insights into host-symbiont interactions, enable the cultivation of potential isolates in laboratory settings, and pave the way for obtaining high-quality genomes from these isolates. Conventional approaches limited to metagenomics cannot offer such insights.

The use of high-throughput sequencing enabled to finely characterize bacterial and protist diversity across all samples. While it was clear that bacterial communities were strongly influenced by various physicochemical parameters, such as sample origin, pH, and salinity/conductivity (Fig. 3F), no definitive impact of those parameters was observed regarding *Ca*. Babelota presence and abundance. Nonetheless, certain trends in abundance were highlighted, such as lower abundance in the spring season and a high and stable presence in forest soils and salt marshes (Fig. 3B).

The omnipresence of *Ca*. Babelota in environmental samples was confirmed using both qPCR and high-throughput sequencing, being present in 85 out of 97 screened samples. Overall, our study corroborates that *Ca*. Babelota can colonize a wide array of niches, as witnessed by previous *in silico* surveys (14). Notably, though highly prevalent, *Ca*. Babelota were systematically detected only in forest soil and salt marshes. Additionally, salt marsh samples exhibited the highest abundance in *Ca*. Babelota, compared to all other samples in this study. Such environmental category warrants further investigation, especially with the use of other sequencing techniques such as whole metagenomic analyses, to recover more comprehensive genomic information on this phylum.

A more detailed analysis of *Ca*. Babelota diversity highlighted that while present in all analyzed environments, the observed diversity was strongly specific to the sample origin, with few ASV found in multiple environments (Fig. 4C). In fact, only two ASV were found in three different environmental categories; both classified under the *Vermiphilaceae* family. Among those, a single ASV was found in both fresh and marine samples. This ASV is closely related to *V. pyriformis*; its wide distribution could have contributed to its successful and multiple recovery (19, 21). Additionally, phylogenetic analyses also indicated that this family stood out as the most diverse and widely distributed within the phylum (Fig. 5). In comparison, other known families were much more restricted, both in terms of ASV representation and environmental tropism. The phylogenetic analysis of *Ca*. Babelota ASV highlighted a complete coverage of all known *Ca*. Babelota clades, including those identified solely through 16S rRNA sequences, such as the F38 clade (15) and a putative novel class within the *Ca*. Babelota (Babelota class II). In this way, the sequencing effort produced for this study seemingly provides an exhaustive picture of the diversity encompassed by the Ca. Babelota phylum. The integration of environmental origins associated with ASV contributes to further characterize and gather new (biological and genomic) data for those poorly described clades. This could be especially relevant for *Ca*. Babelota class II, which seemingly represents a class-level

clade within the *Ca*. Babelota but completely lacks genomic data. Our study indicated that this clade was identified in all environments but soil. Such findings could, thus, guide future metagenomic sequencing effort aimed at recovering complete genomes for representatives of this clade.

We simultaneously sequenced protist diversity across all environmental samples, using a combination of adapted universal primers, coupled with Oxford Nanopore sequencing. This approach enabled us to exploit the full length of amplicon generated by the selected primers, eliminating the need for nested PCR, which is required for adapting this assay to short read sequencing (56).

Protist communities, represented by 22 phylum-level clades, were significantly impacted by temperature and pH. Also, our findings indicated that several protist phyla played a crucial role in shaping bacterial community composition across various environments (Fig. 6D). Many of these protist phyla are heterotrophs, thus, likely highlighting the impact of protist predation on bacterial community. Notably, a significant impact of Dinoflagellata (Alveolata) and Gyrista (Stramenopiles) was observed, toward marine bacterial communities. These results align with previous studies assessing the impact of protist grazing on bacterial communities in similar biomes (57).

Given our hypothesis that protist hosts *Ca*. Babelota, cross-kingdom cooccurrences analyses provide clues regarding protist taxa that could interact with *Ca*. Babelota. This approach may be particularly relevant in cases of stable, non-parasitic associations. The exploration of positive cooccurrences involving *Ca*. Babelota revealed that most cooccurring protists were heterotrophs. Notably, *Acanthamoeba*, a known host for *Ca*. Babelota, was the most frequently cooccurring protist in our analyses. Another known symbiotic association was retrieved in our cooccurrence analyses, involving the free-living amoeba *Vermamoeba vermiformis* and its endosymbiont *Vermiphilus pyriformis* (Fig. 6E) (19). The (frequent) recovery of known host-symbiont pairs in our analyses provides an internal validation of this approach. A total of 5 phylum-level protist clades, represented by 31 genera, were found to positively cooccur with *Ca*. Babelota. Those identified genera constitute as many putative hosts and further studies could build on these findings to recover additional isolates. Among those, *Prorocentrum* stood out as one of the most frequently encountered cooccurrence. This mixotrophic dinoflagellate was repeatedly described in association with several bacteria as part of the phycosphere, even showing symbiotic interactions with bacteria of the *Roseobacter* genus (58–60). This finding corroborates previous observations of *Ca*. Babelota being identified in associations with dinoflagellates (61, 62). Given the involvement of *Prorocentrum* in harmful algal blooms, further investigations into this putative association could provide clues for better understanding the dynamics of such events (63).

Most of the identified protists are known to be heterotrophic though few are autotrophic raising questions about the nature of these co-occurrences. Among the identified autotrophs, five diatoms genera were detected. Although no intracellular bacteria were described within these diatoms, their co-occurrence with *Ca*. Babelota could suggest its presence (by itself or within unidentified hosts) in the diatoms phycosphere (64).

Among algae, three genera cooccurred with *Ca*. Babelota. Among those, a single genus within the golden algae was identified, *Chrysosaccus* sp. This poorly known genus falls within the Chrysophyceae class, which bears the genus *Spumella*, a known host for *Ca*. Babelota (20). This cooccurrence is thus prompting for more investigations to explore the possibility that *Chrysosaccus* (and more broadly Chrysophyceae) could harbor *Ca*. Babelota bacteria. Those algae were found to positively interact with species of the UBA12409 family, for which no cultivable representative is available as of 2024. Such analyses may, thus, constitute pertinent leads for recovering additional, phylogenetically diverse *Ca*. Babelota.

In conclusion, our study confirms that *Ca*. Babelota is among the most frequently occurring rare phyla. Their presence is likely to be linked and affected by protists

though the true host range for those bacteria remains to be defined. Protist enrichments from environmental samples constitute a promising approach for recovering additional isolates of this elusive phylum. The combination of benchwork and statistical, inference-based analyses allowed for suggesting the numerous putative interactions taking place between various *Ca*. Babelota and protists, paving the way for finding future isolation of novel *Ca*. Babelota strains in their native hosts. Further analyses should focus on recovering those isolates using cell sorting techniques, while generating whole (meta)genomes to gain further insight into the *Ca*. Babelota lifestyle.

## ACKNOWLEDGMENTS

We thank Jérôme Labanowski from the Institut de Chimie des Milieux et Matériaux de Poitiers for his help and expertise on sampling. We thank Anne Cantereau-Becq from ImageUP (University of Poitiers) for the help with confocal microscopy observations. We also thank Sébastien Rossignol from Marais salant de Mornac sur Seudre for granting us access to his salt marsh and allowing sampling of water and associated sediments. We acknowledge the Ebioinfo facility at the UMR CNRS 7267 at the University of Poitiers for providing the computing infrastructure.

This work was funded by the Agence Nationale de la Recherche, project ANR-21-CE02-0001 DEPEND.

## AUTHOR AFFILIATION

[1]Laboratoire Écologie and Biologie des Interactions, UMR CNRS 7267, Université de Poitiers, Poitiers, Nouvelle-Aquitaine, France

## AUTHOR ORCIDs

Louis Weisse  http://orcid.org/0000-0003-3320-4686
Lucile Martin  http://orcid.org/0009-0007-8714-6465
Yann Héchard  http://orcid.org/0000-0002-1224-4059
Vincent Delafont  http://orcid.org/0000-0003-1111-2916

## FUNDING

| Funder | Grant(s) | Author(s) |
| --- | --- | --- |
| Agence Nationale de la Recherche | ANR-21-CE02-0001 | Vincent Delafont |

## AUTHOR CONTRIBUTIONS

Louis Weisse, Data curation, Formal analysis, Investigation, Methodology, Validation, Visualization, Writing – original draft | Lucile Martin, Data curation, Formal analysis, Methodology, Visualization | Bouziane Moumen, Data curation, Formal analysis, Methodology | Yann Héchard, Conceptualization, Formal analysis, Validation, Writing – original draft, Writing – review and editing | Vincent Delafont, Conceptualization, Data curation, Formal analysis, Funding acquisition, Investigation, Methodology, Supervision, Validation, Visualization, Writing – original draft, Writing – review and editing

## DATA AVAILABILITY

All raw data have been deposited in the Sequence Read Archive (SRA) under project number PRJNA1171726.

## ADDITIONAL FILES

The following material is available online.

## Supplemental Material

**File S1 (mSystems00261-25-s0001.docx).** Details on primer and probes design and validation.

**Figure S1 (mSystems00261-25-s0002.pdf).** Mean abundances of *Ca*. Babelota detected by quantitative PCR, per sample.

**Figure S2 (mSystems00261-25-s0003.pdf).** *Ca*. Babelota can be maintained in culture through protist enrichments and are systematically intracellular.

**Figure S3 (mSystems00261-25-s0004.pdf).** Rank abundance of all identified *Ca*. Babelota ASV.

**Table S1 (mSystems00261-25-s0005.xlsx).** Full list of samples analyzed in the present study, along with associated physicochemical parameters.

**Table S2 (mSystems00261-25-s0006.xlsx).** Relative abundance of protist genera, for protists enrichments that are positive for *Ca*. Babelota presence.

**Table S3 (mSystems00261-25-s0007.xlsx).** Relative abundance of protist genera, for each analyzed sample.

**Table S4 (mSystems00261-25-s0008.xlsx).** Positive cooccurrences quantitative and statistical data.

## Open Peer Review

**PEER REVIEW HISTORY (review-history.pdf).** An accounting of the reviewer comments and feedback.

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
