## [Reviewer comments · mSystems]

Environmental diversity of *Candidatus* Babelota and their relationships with protists

Louis Weisse, Lucile Martin, Bouziane Moumen, Yann Hechard, and Vincent Delafont

Corresponding Author(s): Vincent Delafont, Universite de Poitiers

Review Timeline:

Submission Date:	February 21, 2025
Editorial Decision:	April 19, 2025
Revision Received:	April 25, 2025
Accepted:	May 1, 2025

Editor: Rosie Alegado

Reviewer(s): Disclosure of reviewer identity is with reference to reviewer comments included in decision letter(s). The following individuals involved in review of your submission have agreed to reveal their identity: Qinglin Chen (Reviewer #2)

Transaction Report:

DOI: <https://doi.org/10.1128/msystems.00261-25>

Re: mSystems00261-25 (**Environmental diversity of *Candidatus* Babelota and their relationships with protists**)

Dear Dr. Vincent Delafont:

Revision Guidelines

Sincerely,
Rosie Alegado
Editor
mSystems

Reviewer #1 (Comments for the Author):

Overall, the study presents a well-structured and valuable exploration of the ecological diversity of an uncultivable phylum of intracellular bacteria *Ca. Babelota* and its interactions with protists. The combination of high-throughput sequencing, protist enrichment cultures, and fluorescence in situ hybridization (FISH) demonstrates a strong effort to overcome the challenges of studying these elusive bacteria. The manuscript effectively contextualizes the significance of understanding symbiotic

interactions between protists and uncultivable bacteria. This manuscript is a valuable contribution to microbial ecology, with great potential for advancing our understanding of symbiotic relationships between protists and intracellular bacteria.

The work and the manuscript are robust and clearly detailed, I only have minor critics and demands to improve few points in the text and some figures.

Minor points:

- Line 22-24: Please rephrase the sentence, use of "further" is atypical.
- Line 41: Please rephrase the sentence, the use of "upon" is somewhat awkward.
- Line 61: Please rephrase the sentence, use of "further" is atypical, replace by "majority".
- Line 71: Please replace "probe" by "investigate".
- Lines 308-310: What do you mean by "potential host", does it suggest that other multispecies cooccurrences are not potential hosts ?
- Figure 4C: please improve the representation and provide a scale for the symbol size related to the ASV counts.
- Figure 5: please improve the visualisation of ASV environmental origin, the symbols are too small.
- Figure 6E: The figure doesn't illustrate well quantitative data and cooccurrence frequency. Please provide the data of the ASVs sequences and taxonomic affiliation that cooccur in an additional table. In addition, statistical analysis of the co-occurrence networks could be better detailed in the main text or the material section.

Reviewer #2 (Comments for the Author):

Summary: The manuscript presents a comprehensive investigation of the environmental distribution, diversity, and host associations of *Candidatus Babelota*, a poorly understood phylum of intracellular bacteria. The authors combine environmental sampling, protist enrichment, molecular detection, and co-occurrence analysis to explore the ecological patterns and potential protist hosts of *Ca. Babelota*. Despite limited cultivation success, the study shows that protist enrichment is a promising approach for recovering *Ca. Babelota* and observing host associations. The identification of a putative novel class within the phylum underscores the need for further genomic exploration. Overall, the work advances understanding of *Ca. Babelota* ecology and provides a foundation for future efforts to isolate and characterize this elusive lineage. I believe this study is timely and contributes to the understanding of microbial dark matter and protist-bacterial symbioses. However, several important concerns must be addressed before the manuscript can be considered for publication.

Candidatus Babelota is a phylum of strictly intracellular bacteria, does protists is their only hosts?

Through protist enrichment and fluorescence in situ hybridization (FISH), *Ca. Babelota* cells were directly visualized within protist hosts, confirming their intracellular lifestyle. A probe specifically targeting *Ca. Babelota* was designed and used in this study, how to evaluate the specificity of the probe?

This study demonstrates that enriching protist hosts is a promising approach for recovering *Candidatus Babelota*, which is indeed a very interesting and valuable strategy. However, it appears that this method may be selective, as it predominantly enriches for specific *Ca. Babelota* clades—namely those associated with protist hosts that can be cultivated under laboratory conditions. This suggests that while protist enrichment is effective, it may not fully capture the broader diversity of *Ca. Babelota*, especially lineages associated with uncultivable or less-studied protist taxa.

While the enrichment of protist hosts offers a valuable route to recover *Candidatus Babelota*, it raises the question of necessity: if metagenomic approaches already allow for the detection and genomic reconstruction of *Ca. Babelota*, why is enrichment still required? I believe the authors should clarify what unique advantages this method offers beyond conventional metagenomic analysis—such as enabling physical observation of host-symbiont interactions, improving the likelihood of obtaining high-quality genomes, or facilitating downstream experimental studies. Explicitly addressing this distinction would help justify the added value of the enrichment-based approach.

Summary: The manuscript presents a comprehensive investigation of the environmental distribution, diversity, and host associations of *Candidatus Babelota*, a poorly understood phylum of intracellular bacteria. The authors combine environmental sampling, protist enrichment, molecular detection, and co-occurrence analysis to explore the ecological patterns and potential protist hosts of *Ca. Babelota*. Despite limited cultivation success, the study shows that protist enrichment is a promising approach for recovering *Ca. Babelota* and observing host associations. The identification of a putative novel class within the phylum underscores the need for further genomic exploration. Overall, the work advances understanding of *Ca. Babelota* ecology and provides a foundation for future efforts to isolate and characterize this elusive lineage. I believe this study is timely and contributes to the understanding of microbial dark matter and protist-bacterial symbioses. However, several important concerns must be addressed before the manuscript can be considered for publication.

Candidatus Babelota is a phylum of strictly intracellular bacteria, does protists is their only hosts?

Through protist enrichment and fluorescence in situ hybridization (FISH), *Ca. Babelota* cells were directly visualized within protist hosts, confirming their intracellular lifestyle. A probe specifically targeting *Ca. Babelota* was designed and used in this study, how to evaluate the specificity of the probe?

This study demonstrates that enriching protist hosts is a promising approach for recovering *Candidatus Babelota*, which is indeed a very interesting and valuable strategy. However, it appears that this method may be selective, as it predominantly enriches for specific *Ca. Babelota* clades—namely those associated with protist hosts that can be cultivated under laboratory conditions. This suggests that while protist enrichment is effective, it may not fully capture the broader diversity of *Ca. Babelota*, especially lineages associated with uncultivable or less-studied protist taxa.

While the enrichment of protist hosts offers a valuable route to recover *Candidatus Babelota*, it raises the question of necessity: if metagenomic approaches already allow for the detection and genomic reconstruction of *Ca. Babelota*, why is enrichment still required? I believe the authors should clarify what unique advantages this method offers beyond conventional metagenomic analysis—such as enabling physical observation of host-symbiont interactions, improving the likelihood of obtaining high-quality genomes, or facilitating downstream experimental studies. Explicitly addressing this distinction would help justify the added value of the enrichment-based approach.

Dear Dr. Vincent Delafont:

Dear Dr. Rosie Alegado, thank you very much for your feedback on our submitted manuscript. We are thankful to your and the reviewer's comments and strived to take those into account for improving the manuscript. Please find below, as recommend by the guidelines, a point-by-point answer to all questions raised by the reviewers, hoping we have convincingly addressed it.

Revision Guidelines

- Upload point-by-point responses to the issues raised by the reviewers in a file named "Response to Reviewers," NOT in your cover letter.
- Upload a compare copy of the manuscript (without figures) as a "Marked-Up Manuscript" file.
- Upload a clean .DOC/.DOCX version of the revised manuscript and remove the previous version.
- Each figure must be uploaded as a separate, editable, high-resolution file (TIFF or EPS preferred), and any multipanel figures must be assembled into one file.
- Any [supplemental material](https://journals.asm.org/writing-your-paper#supplemental-material) intended for posting by ASM should be uploaded with their legends separate from the main manuscript. You can combine all supplemental material into one file (preferred) or split it into a maximum of 10 files with all associated legends included.

For complete guidelines on revision requirements, see our [Submission and Review Process](https://journals.asm.org/journal/msystems/submission-review-process) webpage. Submission of a paper that does not conform to guidelines may delay acceptance of your manuscript.

Publication Fees: For information on publication fees and which article types are subject to charges, visit our [website](https://journals.asm.org/publication-fees). If your manuscript is accepted for publication and any fees apply, you will be contacted separately about payment during the production process; please follow the instructions in that e-mail. Arrangements for payment must be made before your article is published.

ASM Membership: Corresponding authors may [join or renew ASM membership](https://www.asm.org/membership) to obtain discounts on publication fees. Need to upgrade your membership level? Please contact Customer Service at Service@asmusa.org.

The ASM Journals program strives for constant improvement in our submission and publication process. Please tell us how we can improve your experience by taking this quick survey.

<https://www.surveymonkey.com/r/ASMJournalAuthors> Author Survey

Sincerely,
Rosie Alegado
Editor
mSystems

Reviewer #1 (Comments for the Author):

Overall, the study presents a well-structured and valuable exploration of the ecological diversity of an uncultivable phylum of intracellular bacteria *Ca. Babelota* and its interactions with protists. The combination of high-throughput sequencing, protist enrichment cultures, and fluorescence in situ hybridization (FISH) demonstrates a strong effort to overcome the challenges of studying these elusive bacteria. The manuscript effectively contextualizes the significance of understanding symbiotic interactions between protists and uncultivable bacteria. This manuscript is a valuable contribution to microbial ecology, with great potential for advancing our understanding of symbiotic relationships between protists and intracellular bacteria.

The work and the manuscript are robust and clearly detailed, I only have minor critics and demands to improve few points in the text and some figures.

We thank the reviewer for the valuable input and the comments are much appreciated.

Minor points:

- Line 22-24: Please rephrase the sentence, use of "further" is atypical.

The sentence was modified to avoid using 'further'

- Line 41: Please rephrase the sentence, the use of "upon" is somewhat awkward.

The sentence was modified accordingly

- Line 61: Please rephrase the sentence, use of "further" is atypical, replace by "majority".

The sentence was modified

- Line 71: Please replace "probe" by "investigate".

This modification was implemented in the manuscript

- Lines 308-310: What do you mean by "potential host", does it suggest that other multispecies cooccurrences are not potential hosts ?

We wanted to mean that monospecies cooccurrences might provide more precise information regarding the identification of putative natural associations. We slightly rephrased this section to make it clearer to the reader

- Figure 4C: please improve the representation and provide a scale for the symbol size related to the ASV counts.

We have tried to improve the figure and implemented a scale for better interpreting the sequence number for each ASV.

- Figure 5: please improve the visualisation of ASV environmental origin, the symbols are too small.

We thank the reviewer for the suggestion. We have tried to make symbols bigger, but it leads to massive overlaps, thus impairing the figure readability. We aimed to make this figure 2/3 of a page in final format, and think that using this size, the symbol will be readable. Moreover, the electronic version should allow to zoom in to a sufficient level to focus on any part of the figure if needs be. For these reasons we would respectfully ask to leave the figure as is.

- Figure 6E: The figure doesn't illustrate well quantitative data and cooccurrence frequency. Please provide the data of the ASVs sequences and taxonomic affiliation that cooccur in an additional table. In addition, statistical analysis of the co-occurrence networks could be better detailed in the main text or the material section.

We thank the reviewer for the pertinent comment and modified the figure 6E according to this suggestion. We believe it greatly improved the visualization of cooccurrences. Added to that, we provide a supplementary table (#4) that compile of the numeric data used and generated using the coocurr package, enabling the reader to go deeper in the quantitative data used for building the network. Additional details were also given in the methods section on the values considered for building the network, along with the associated statistical significance.

Reviewer #2 (Comments for the Author):

Summary: The manuscript presents a comprehensive investigation of the environmental distribution, diversity, and host associations of Candidatus Babelota, a poorly understood phylum of intracellular bacteria. The authors combine environmental sampling, protist enrichment, molecular detection, and co-occurrence analysis to explore the ecological patterns and potential protist hosts of Ca. Babelota. Despite limited cultivation success, the study shows that protist enrichment is a promising approach for recovering Ca. Babelota and observing host associations. The identification of a putative novel class within the phylum underscores the need for further genomic exploration. Overall, the work advances understanding of Ca. Babelota ecology and provides a foundation for future efforts to isolate and characterize this elusive lineage. I believe this study is timely and contributes to the understanding of microbial dark matter and protist-bacterial symbioses. However, several important concerns must be addressed before the manuscript can be considered for publication.

We are deeply thankful to the reviewer for the pertinent comments. Please find below our answers to the questions that we are striving to address.

Candidatus Babelota is a phylum of strictly intracellular bacteria, does protists is their only hosts? Protists are the only known hosts for the various, phylogenetically unrelated, Ca. Babelota isolates known to this day. This was the point encouraging us to hypothesize that this phylum may preferentially (or even exclusively?) infect protists. However, at this point, we cannot exclude the fact that other hosts may exist for these bacteria (cf. Discussion on line 330-332). Despite that, we believe our study greatly expands our knowledge on putative protist host harboring Babelota.

Through protist enrichment and fluorescence in situ hybridization (FISH), Ca. Babelota cells were directly visualized within protist hosts, confirming their intracellular lifestyle. A probe specifically targeting Ca. Babelota was designed and used in this study, how to evaluate the specificity of the probe?

We understand the reviewer's point and would like to direct his/her attention to the supplementary file 1, providing additional data on probe design and validation. The probe specificity was tested in silico, and probe binding was also evaluated in our lab using both *Vermiphilus pyriformis* and *Babela massiliensis*, available in culture within amoebae.

This study demonstrates that enriching protist hosts is a promising approach for recovering Candidatus Babelota, which is indeed a very interesting and valuable strategy. However, it appears that this method may be selective, as it predominantly enriches for specific Ca. Babelota clades—namely those associated with protist hosts that can be cultivated under laboratory conditions. This suggests that while protist enrichment is effective, it may not fully capture the broader diversity of Ca. Babelota, especially lineages associated with uncultivable or less-studied protist taxa.

We agree with the reviewer and have tried to briefly discuss this point on lines 330-332 in the original manuscript. We have slightly expanded this section to better consider this point

While the enrichment of protist hosts offers a valuable route to recover Candidatus Babelota, it raises the question of necessity: if metagenomic approaches already allow for the detection and genomic

reconstruction of *Ca. Babelota*, why is enrichment still required? I believe the authors should clarify what unique advantages this method offers beyond conventional metagenomic analysis-such as enabling physical observation of host-symbiont interactions, improving the likelihood of obtaining high-quality genomes, or facilitating downstream experimental studies. Explicitly addressing this distinction would help justify the added value of the enrichment-based approach.

We thank the reviewer for this comment. Indeed, getting access to biological isolates through cultivation (regardless of the approach) open many more doors in terms of characterization of host-bacterium interactions and constitutes one of the few ways to get insights into the cell biology of microorganisms. We believe it was especially important for such a poorly understood phylum, to make this extra effort, even though metagenomics data can also provide a lot of information. We have tried to better emphasize on the added value of such an approach in the manuscript (line 381-385 in the marked_up version), taking inspiration from the reviewer's comment.

Re: mSystems00261-25R1 (**Environmental diversity of *Candidatus* Babelota and their relationships with protists**)

Dear Dr. Vincent Delafont:

Your manuscript has been accepted, and I am forwarding it to the ASM production staff for publication. Your paper will first be checked to make sure all elements meet the technical requirements. ASM staff will contact you if anything needs to be revised before copyediting and production can begin. Otherwise, you will be notified when your proofs are ready to be viewed.

Sincerely,
Rosie Alegado
Editor
mSystems